# Treat to Target (Drug-Free) Inactive Disease in JIA: To What Extent Is This Possible?

**DOI:** 10.3390/jcm11195674

**Published:** 2022-09-26

**Authors:** Athimalaipet V. Ramanan, Anne M. Sage

**Affiliations:** 1Bristol Royal Hospital for Children, Bristol BS8 1QU, UK; 2Translational Health Sciences, University of Bristol, Bristol BS8 1TS, UK; 3Department of Rheumatology, Perth Children’s Hospital, 15 Hospital Avenue, Nedlands, WA 6009, Australia

**Keywords:** JIA, T2T

## Abstract

**Background**: Treat to target (T2T) is a strategy that has been increasingly employed in the management of several chronic diseases, with demonstrated improved outcomes. The use of T2T in juvenile idiopathic arthritis (JIA), a common rheumatic disease of childhood, is still in its infancy, and the feasibility of its use in attaining drug-free clinical remission is unclear. **Aims**: We aim to explore the current literature of the use of T2T in JIA, and to review the potential benefits and limitations of this approach in regard to this chronic disease. **Sources**: A comprehensive PubMed search was conducted using relevant keywords, with full text articles in English included in the review. **Content**: T2T is an appealing strategy for improving outcomes of pediatric rheumatic diseases given the limited availability of therapeutics and potential cumulative effects of long-term immunosuppression. The application in a cohort of children, however, is limited by heterogeneity of disease, availability of high-quality evidence, and patient and parental preferences. Unlike adult rheumatoid arthritis, the ‘window of opportunity’ has not been definitively demonstrated in large scale trials, and although early studies of T2T in JIA have been favorable, the timing and means of escalation (especially with regard to biologics) need clarification. **Implications**: This review outlines several issues of implementing T2T in JIA, including the important extra-articular manifestations of disease and non-pharmacological management, that should be considered in future consensus guidelines.

## 1. Introduction

Treat to target (T2T) is a strategy which uses a target endpoint to guide therapeutic decisions in the management of chronic diseases. Key components of T2T are active patient participation, creation of clear goals, regular reviews, and escalation of therapy until goals are met. 

T2T was first adopted by rheumatologists in the management of rheumatoid arthritis (RA), with recommendations published from an expert international task force in 2010 [1]. The basis for this consensus recognized that advancements in understanding, disease activity measurement tools and novel treatment options made clinical remission for many a realistic target. Early elimination of inflammation was the foundation of these recommendations in preventing joint disease and physical disability, both of which have been correlated with poorer quality of life and premature death. 

This T2T principle has been a key concept in driving better outcomes for adults with rheumatoid arthritis, as evidenced by a wealth of research subsequent to these recommendations [2]. It is now considered the standard of care for adults with RA [3,4,5] and has been adapted for other rheumatological disorders, such as ankylosing spondylitis [6], psoriasis [6], systemic lupus erythematosus [7], and more recently in knee osteoarthritis [8] and osteoporosis [9]. In RA, the utilisation of the T2T strategy has been demonstrated to be more important than the therapeutic agent itself [10,11,12,13], with one small meta-analysis suggesting that protocol-driven tight control may have additional benefit [14]. It has also been shown to be efficacious even in established disease [15]. 

Despite internationally accepted guidelines, which have undergone several revisions to address consumer and physician concerns, and demonstrated benefit over standard management, T2T for RA is not uniformly adhered to in clinical practice [15]. Some barriers include confidence in, and heterogeneity of composite disease activity measures; consultation time pressure; patient suboptimal adherence to, reluctance to change and side effects of treatment; and physician understanding of T2T guidelines; amongst others [15,16].

For this narrative review, we have conducted a comprehensive PubMed search of full text English articles using a search strategy of relevant keywords, and hence discuss the use of T2T in management of JIA. 

JIA Quick Facts [17]
Heterogenous group of chronic arthritidies (>6 weeks duration), onset before age of 16 and at the exclusion of other diagnoses Incidence between 1.6 to 23 per 100,000, prevalence 3.8 to 400 per 100,000 children in Europe Female > male Oligoarthritis most common subtype

## 2. Why T2T for Juvenile Idiopathic Arthritis (JIA)?

Initial and ongoing treatment of JIA differs depending on subtype, with consensus guidelines largely based upon low or moderate level evidence [18,19]. NSAIDs and intra-articular corticosteroids are mainstay of initial treatment of oligoarthritis, with escalation to DMARDs and biologic DMARDs in case of progressive or recalcitrant disease, or extraarticular disease such as uveitis [18]. Conversely, early DMARD use is recommended in polyarthritis, and systemic corticosteroid use is more commonplace [19]. Treatment differs again in the systemic onset JIA cohort, a predominately autoinflammatory condition, with systemic corticosteroids and early biologic DMARD use common, especially in the case of macrophage activation syndrome [19]. 

Given inherent difficulties in conducting interventional research and limited therapeutic options in the pediatric population, advances in management of childhood rheumatic diseases often follow that of similar adult conditions. Based on the success of targeted therapy in RA, a taskforce was established which set recommendations for T2T in JIA [20]. Like in RA, ‘*abrogation of inflammation*’ was the cornerstone of these consensus principles, with addition of other important pediatric considerations of avoidance of long-term glucocorticoids, optimization of growth and development and ensuring absence of extraarticular manifestations of JIA, such as uveitis [20].

Clinical remission was again the proposed standard in this consensus, which would routinely be the aim of treating pediatric rheumatologists globally. Wallace et al. (2004) [21] defined clinical remission in JIA as that on medication (inactive disease on medication for a period of six months) and off medication (inactive disease off all medications for a period of 12 months); criteria that are widely utilized in outcome measurement today. The manner in which this is achieved, however, may differ greatly between clinicians, and the trade-off between medications and disease control variably interpreted. The T2T approach may provide clarity not only to the physicians, but also the parents and patients, and allow for a truly collaborative approach. 

Compared to the historic cohort, the outcomes for young people today with JIA have markedly improved. Inability to achieve and/or maintain clinical remission, either on or off medications, remains a concern for a significant proportion of this patient group, however, particularly in the rheumatoid factor (RF) positive subgroup. Given attainment of clinical remission at least once in the disease process has been correlated with improved long term outcomes in JIA [22], the T2T approach which uses this as the intended target should similarly result in better prognosis.

## 3. How Realistic Is Clinical Remission (Drug Free) in JIA

Unlike RA in adults, JIA is much more heterogenous. Disease course, treatment approach and outcomes (both articular and extra-articular) vary vastly between, and indeed within ILAR subgroups [23]. Whilst in many cases, parents will be counselled at presentation that their child would be expected to finish school in drug free clinical remission, the prognosis remains far more guarded for others. 

JIA inception cohorts, such as Research in Arthritis in Canadian Children emphasizing Outcomes (ReACCh-Out) in Canada and Childhood Arthritis Prospective Study (CAPS) in the UK, were created to investigate factors associated with outcomes, and help guide clinicians in their decision making. 

Patient outcomes on conventional therapies in the ReACCh-Out cohort (recruited between 2005 and 2010) were published in 2015 [24]. Of the 1104 analyzed, 80% were able to reach inactive disease (based on the criteria by Wallace et al. [21] with additional modifications) at a median of 13 months post diagnosis across all subtypes. Hence, 178 children (16.1% of the cohort) achieved disease remission (defined as 12 months of inactive disease off all medications), with a higher cumulative probability in the oligoarthritis subgroup (57% over five years). Notably, none of the RF positive polyarticular group were able to reach disease remission, with only 8% cumulative probability of reaching inactive disease by 5 years. Further work based on this cohort data resulted in the development of models to predict those who had a low chance of remission (above JIA subtype alone) [25] and those who would have a severe disease course [26], which although they are not perfect, may help to set initial treatment goals as part of T2T. 

Disease trajectories and variables at presentation and over time of children in the CAPS cohort were reported in 2021 [27]. This study demonstrated that there were ‘clusters’ within this cohort of disease severity and longitudinal course, and whilst activity at presentation was congruent for many with pattern of disease over time, this was not uniform. Over a three year follow up period, low disease activity and remission were seen in 66% of the cohort, primarily oligoarthritis subgroups, although treatment was not directly reported. 

Bava and colleagues (2019) [28] also looked to identify factors associated with inability to reach inactive disease in their retrospective review of 375 patients with JIA who were treated with methotrexate as the sole DMARD. Hence, 61% of patients were able to reach inactive disease as per Wallace criteria (or caring physician’s assessment where all components of criteria were unavailable) at a median of 1.7 years post initiation of methotrexate. Factors predictive of not achieving inactive disease included enthesitis-related and systemic arthritis subtypes; ANA negative status; higher ESR and CRP; male and older at disease onset; with an equal number in the polyarticular subgroup reaching inactive disease as did not. 

## 4. T2T for JIA in the Literature

Unlike the RA equivalent, there were no randomized controlled trials in JIA that directly compared targeted to standard therapy on which the taskforce was able to base the JIA T2T recommendations. There were, however, two randomized studies of children with polyarthritis which used early and aggressive therapies with set review points, at which time change (predominately, escalation) of therapies was mandated if predetermined improvement goal hadn’t been reached, similar to the T2T strategy [29,30]. 

Subsequent to the recommendations, several centers adopted this strategy, with some groups publishing their experience in recent years. 

Hissink Muller and colleagues (2019) [31] looked at T2T (drug-free inactive disease) in comparing three treatment regimens (stepwise conventional DMARDs, combination DMARD and glucocorticoids, combination methotrexate and etanercept) for young people who were DMARD-naïve with psoriatic, oligo- and polyarthritis (RF-ve). Unlike the known epidemiology of JIA [17], most were in the polyarticular subgroup. Regardless of treatment initiated, inactive disease was achieved by approximately 71% of all children by end of two years, 39% of whom were drug free. Noting the short period of follow up, these results suggest that the early implementation of biologic agents or addition of prednisolone were not superior to a step wise escalation of conventional DMARDs. 

Klein et al. (2020) [32], compared T2T to standard therapy (based on BIKER registry data) in children with either polyarthritis (seronegative and positive) or extended oligoarthritis. Whilst more patients in the T2T group reached remission and minimal disease activity (as per JADAS), more children received systemic steroids and biologic therapy than in the historic cohort (noting likely bias in biologic use in more recent years), although less received systemic steroids after 12 months. Importantly, half the group reached inactive disease with methotrexate as the only disease modifying agent. 

More recently, at T2T approach was demonstrated to be successful in decreasing disease activity, increasing function and decreasing pain in children with both new and established polyarticular JIA [33]. This study used clinical decision support (CDS) algorithms to guide treatment based on disease activity, current medication, and dose, in an effort to decrease physician variability in therapeutic choice. The advantage of CDS use above unstructured T2T was not addressed, nor the T2T compared against a standard approach. However, in two thirds of the 213 visits, the target of ‘not active’ was met [33].

The T2T approach has also been evaluated in the management of systemic JIA. In a small cohort, Vastert and colleagues [34] demonstrated that the early instigation of anakinra with frequent reviews resulted in 85% attaining clinically active disease at 12 months post diagnosis. In contrast to many others, this study did demonstrate the superiority of early biologic implementation (when compared to results from other studies where anakinra was introduced later in the disease course), supporting the notion of ‘window of opportunity’ in the management of inflammatory rheumatic diseases. Given anakinra is an expensive medication, beyond the reach of many on continents, such as Asia and Africa, the verification of these results with larger scale interventional trials should be undertaken. 

## 5. Challenges with T2T in JIA 

Implementation of the T2T strategy in the management of young people with JIA requires consideration of many other factors, which may not be as apparent in equivalent adult diseases. 

Across the limited literature of T2T in JIA, protocol violations and reluctance to participate in the study were notable. In pediatric medicine, the parent or caregiver is most often making treatment decisions on behalf of the young person. Family reluctance to participate in trials, intensify treatment and /or use medications that are off label all may limit the efficacy of this strategy, and reinforce the importance of shared decision making. Although the success of the consumer version of recommendations for RA has not been investigated, a similar undertaking for T2T guidelines in JIA may help to improve patient and parent understanding in creating shared goals. 

Availability of therapeutics, such as biologic DMARD’s, is far more limited in pediatric medicine compared to that of adults, with off-label and unlicensed use common, and delay in approval for many [35]. Additionally, the expense associated with such therapies makes them out of reach to many around the globe, especially in countries without universal health care. The pharmacokinetics of drug metabolism in children is also different to adults, with relatively increased clearance of many drugs necessitating increased dose frequency, thereby impacting acceptability to children and their family [35]. Whilst the T2T does not specify therapeutic choice, in the absence of acceptance or availability of newer therapies such as biologics or small molecule agents such as JAK inhibitors (e.g., tofacitnib and baricitinib], other medications used in intensification of treatment to reach the goal may result in serious side effects. Of particular concern are drug toxicities that affect growth and bone health, such as the commonly used glucocorticoids. Limiting the long-term use of systemic steroids was noted in the recommendations for T2T in JIA. 

Extra-articular manifestations of JIA are common, and for many can be more severe than the inflammatory joint disease. 

Uveitis is found in up to 20% of children with JIA [36], with young age of onset, female gender, ANA positivity, oligoarticular subtype and HLA high risk alleles identified as risk factors for development. Potential sight-threatening ocular complications, such as cataracts, glaucoma, band keratopathy, macular oedema, and posterior synechiae, occur in up to half of cases, with visual loss occurring in 10–20% [37]. Whilst many children are well controlled on topical and/or conventional DMARD therapy, there are a subset who have severe and resistant disease. Ocular activity is not always congruent with joint disease, and escalation of treatment with agents such as biologics may be required even in those with quiescent arthritis. Additionally, given topical steroids side effects of ocular hypertension and cataracts have been correlated with increased dose [38], the addition and/or escalation of systemic immunosuppression may be instigated earlier in treatment course. Similarly, dactylitis and enthesitis can be present in the absence of other joint activity and at times resistant to treatment and may require an alteration to the therapeutic regimen. 

Pain, morning stiffness, and fatigue are other factors associated with JIA which can have great impact on function and quality of life. Despite achieving clinical remission based on physician assessment and laboratory results, studies have demonstrated that some young people will still score on the parent/patient VAS [26,27], suggesting that the goal of elimination of inflammation may not be adequate. An example of this would be persistent anterior knee pain following a joint effusion, where biomechanical deficits can lead to ongoing pain long after the inflammatory process has settled. Often associated with persistent pain, the psychological impact of rheumatic diseases in young people cannot be underestimated. In one report, 90% of those aged between 18 and 35 with a rheumatic disease (either diagnosed as a young adult or in childhood) acknowledged mental health difficulties, with only 17% having sought psychological intervention [39]. 

## 6. Future Directions

One of the real challenges of clinical decision making in pediatric rheumatology is the lack of large scale, multicenter, randomized controlled trials. In existing interventional research, the protocols often use an inferior ‘drug withdrawal’ approach rather than a direct ‘head-to-head’ therapeutic comparison, with complex statistical analyses in some finding results that may not be clinically significant to the patient or treating physician. Acknowledging the ethical and logistical difficulties associated with interventional research in young people, ongoing efforts to produce ‘head-to-head’ studies of therapeutic options in JIA will only help to improve outcomes for these patients. 

Although widely presumed, T2T does not necessarily need to be early escalation to expensive medications like biologic DMARD’s and small molecule agents, rather this could mean effective use of cheaper and more accessible therapeutics with a clear focus. Given the vast majority of studies, even in adult RA, do not suggest that early implementation of biologics is clinically and/or cost effective, a step wise escalation of therapies may be superior and more acceptable to patients and families. 

To properly evaluate the effectiveness of the T2T strategy itself, well designed trials that incorporate a T2T approach that is reproducible in everyday clinical care are paramount. Even in the RA patient group, where T2T has been clearly demonstrated to be efficacious over standard approach, it is not uniformly applied, in part due to patient and physician education and time pressures, as listed earlier. 

Thought should be given to make study protocols, and indeed the T2T guidelines, acceptable and realistic for use in a range of countries, not only from a pharmacological perspective but also in the allied health interventions. Many pediatric rheumatologists, predominately from low- and middle-income countries, do not readily have access to services, such as physiotherapy, specialist clinical nurses, and psychology, instead taking on some of these roles themselves. 

There also needs to be a standardization of clinical outcomes across future research, so that results can be directly compared regardless of where the research takes place. The Juvenile Arthritis Disease Activity Score (JADAS) is a composite tool developed to address therapeutic effectiveness both at a patient level and more globally to compare outcomes across centers [40]. This measure and the clinical JADAS (cJADAS), which does not include laboratory inflammatory markers, have been used successfully for T2T in JIA [31,32,33,41], with a similar composite score recently developed for systemic JIA [42]. 

## 7. Conclusions

Young people with JIA today look forward to a future far superior to their predecessors, with vastly improved outcomes even since early in the millennium. The T2T approach, which has been so successful in other chronic diseases, provides an exciting opportunity to improve the care of these children and adolescents, even before newer (and likely more expensive) therapeutic options are available. Increasingly, T2T is being recognized as a key aim, although at this time remains aspirational in the management of JIA given the ambiguity of recommendations and lack of a strong evidence base. Future recommendations should address all aspects of JIA and include expert input from all members of the multidisciplinary team, especially the patient and carers themselves. Lastly, these guidelines should allow the universal practice of T2T, regardless of country, and the use should be realistic even in the case of limited resources. 

Key points
The success of T2T in other chronic diseases presents an exciting opportunity to improve outcomes in young people with JIA Current T2T recommendations provide an important starting point for global implementation in this cohort Efficacy of this approach needs to be assessed with high quality, ‘head to head’ studies, which address all aspects of biopsychosocial model Future recommendations need input from all specialties, and importantly the patient and carer themselves, and need to be globally relevant despite disparities in healthcare

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
