# Peer review of "Treat to Target (Drug-Free) Inactive Disease in JIA: To What Extent Is This Possible?"

_jcm, 2022, doi:10.3390/jcm11195674_

Round 1

Reviewer 1 Report

This is a comprehensive, balanced review on the T2T approach in JIA. The text reads well. My two suggestions:

1. Include a table summarising the key references discussed in the text with focus on the benefits of a T2T approach in JIA treatment.

2. A Table summarising the 'research agenda', i.e. issues that future studies need to tackle.

Author Response

Thank you for your review of our manuscript, and for your comments. Please see responses to your comments below.

This is a comprehensive, balanced review on the T2T approach in JIA. The text reads well. My two suggestions:

1. Include a table summarising the key references discussed in the text with focus on the benefits of a T2T approach in JIA treatment.

Given there are very few articles included in our review, we have highlighted the main references within the reference list itself, which includes the initial consensus guidelines.

2. A Table summarising the 'research agenda', i.e. issues that future studies need to tackle.

To amalgamate a few of reviewer suggestions, we have created a 'key point summary' table, which includes research agenda 

Reviewer 2 Report

Ramanan and Sage's manuscript entitled "Treat to target (drug-free) inactive disease in JIA: to what extent is this possible?" is an interesting topic for the study, where the authors performed an extensive literature search and came to the conclusion that JIA has a greatly beneficial in chronic diseases state, however it is not well utilized in children's disease case. The author's come to the conclusion that JIA has the potential to help children suffering from diseases thereby lowering the patient's care cost.

The strength of the article is that this type of research study can directly help patients suffering from JIA. However, there are weaknesses in the article, which should be properly addressed.

 Line 1-2: Reorganized your title. Try to omit the parentheses, if possible.

Line 4: "Authors and affiliations", are these texts necessary here?

Line 8,9,10: I would place "1" "2" and "3" in the form of superscripts linked to affiliations, which will look nicer.

Line 64-95: Are all these texts fall under the heading "Implications". It is not a good style, you would need to discuss the introduction. At the end of the introduction please do not forget to write what this review focuses on. The highlight of your study should be discussed.

Line 44-67: Please check with journal guidelines whether such headlines "BACKGROUND" "AIMS" "SOURCES" content etc.

Line 75: Check the "2010.(1)" is the right style to reference, here and elsewhere if found.

Line 97: You should reorganize or group all your questions in either the "Result" or "Discussion" section. When you keep moving to the next question does it scientifically or logically follow?

The presentation style does not seem to delve into depth, the article can be greatly improved with the addition of Tables, figures from the experiments, and quantifications.

Line 342: Please make sure that the references follow the journal's guidelines here and elsewhere. For example, see "2013" has been repeated.

Line 250-274: This interpretation without citation of others' work diminishes the worth of content. I would highly recommend including as many citations as possible.
Overall, the review article at the present state needs to follow the journal's guidelines. The article does not reach that depth in explaining the research topics. It is recommended that the authors provide additional tables, figures, schematics, and flow charts. The article should follow scientific patterns on how to write a review, and where to introduce required content.

Author Response

Thank you for reviewing our manuscript, and for your very detailed response. Please see below as responses to your comments. 

Ramanan and Sage's manuscript entitled "Treat to target (drug-free) inactive disease in JIA: to what extent is this possible?" is an interesting topic for the study, where the authors performed an extensive literature search and came to the conclusion that JIA has a greatly beneficial in chronic diseases state, however it is not well utilized in children's disease case. The author's come to the conclusion that JIA has the potential to help children suffering from diseases thereby lowering the patient's care cost.

The strength of the article is that this type of research study can directly help patients suffering from JIA. However, there are weaknesses in the article, which should be properly addressed.

Although it was not clear in the Journal submission process, Professor Ramanan was approached to contribute to a special edition of this journal, given his expertise and research portfolio. The edit given by the journal editor was that of a narrative review, with the title as was submitted. This may go some way in answering some of your concerns about the style of this article, and some of the liberties that have been taken in interpretation and opinion of T2T in JIA management.

We have not followed the traditional format of ‘aims, methods’ etc, rather we have grouped key points about this strategy under headings throughout. The idea of this was to make it easier for the reader to find and follow relevant points within this narrative style.

 Line 1-2: Reorganized your title. Try to omit the parentheses, if possible.
This was the title given to us by the journal, so we have used the wording as directed.

Line 4: "Authors and affiliations", are these texts necessary here?
Line 8,9,10: I would place "1" "2" and "3" in the form of superscripts linked to affiliations, which will look nicer.
These have been edited in the manuscript as suggested. 

Line 64-95: Are all these texts fall under the heading "Implications". It is not a good style, you would need to discuss the introduction. At the end of the introduction please do not forget to write what this review focuses on. The highlight of your study should be discussed.

We have included a paragraph in the introduction which better defines the purpose of this article, as well as a brief description of the methods used.

Line 44-67: Please check with journal guidelines whether such headlines "BACKGROUND" "AIMS" "SOURCES" content etc.
Please see above. 

Line 75: Check the "2010.(1)" is the right style to reference, here and elsewhere if found.

We have used standard Sage Vancouver reference style throughout, a format accepted by this journal. I have written directly to the journal to clarify this point, and will edit accordingly if needed. 

Line 97: You should reorganize or group all your questions in either the "Result" or "Discussion" section. When you keep moving to the next question does it scientifically or logically follow?

The presentation style does not seem to delve into depth, the article can be greatly improved with the addition of Tables, figures from the experiments, and quantifications.

Please see paragraph above, which may answer some of the concerns you have voiced here. We have also added 'quick facts' and 'key points summary' table to introduce JIA to an audience that may not be as familiar with this disease, and clarify main points in the article.   

Line 342: Please make sure that the references follow the journal's guidelines here and elsewhere. For example, see "2013" has been repeated.

Line 250-274: This interpretation without citation of others' work diminishes the worth of content. I would highly recommend including as many citations as possible.
Overall, the review article at the present state needs to follow the journal's guidelines. The article does not reach that depth in explaining the research topics. It is recommended that the authors provide additional tables, figures, schematics, and flow charts. The article should follow scientific patterns on how to write a review, and where to introduce required content.

Please see above paragraph as a response to this comment. Given the narrative review style, some of what is written is opinion based on the current literature. 

Reviewer 3 Report

It is a well structured review showing the scientific knowledge related to T2T aproach in JÄ°A filed.  It is better to present T2T outcomes in a different table. In addition, authors should clarify the methodology for literature selection.

Author Response

Thank you for your review of our manuscript, and valuable comments. Please see our response below

It is a well structured review showing the scientific knowledge related to T2T aproach in JÄ°A filed.  It is better to present T2T outcomes in a different table. In addition, authors should clarify the methodology for literature selection.

Given there are very few articles on T2T in JIA, we have simply highlighted the main articles in the reference section to help readers identify these texts. We have summarised key points in a separate table for the main text. 

We have added a brief explanation of our purpose and methods in the introduction, as suggested. 

Reviewer 4 Report

This is valuable paper on treat to target (drug-free) option in inactive disease in JIA.

The paper is well-written with relevant details and appropriate discussion.

There are some minor issues that should be addressed:

1.       Please add some data regarding main characteristics of JIA. What is the current approach and which are its main disadvantage?

2.       Authors state that it is challenging to achieve remission in JIA patients. Please add details on remission criteria in JIA. When is patient considered to be in remission.

3.       It would be valuable to provide diagram for the treatment of JIA (including T2T option), to make it demonstrative for readers.

4.       There are some papers regarding the JIA and JIA treatment that have been missed. Please re-check.

Author Response

Thank you for your review of our manuscript, and for your valuable comments. Please see responses below

This is valuable paper on treat to target (drug-free) option in inactive disease in JIA.

The paper is well-written with relevant details and appropriate discussion.

There are some minor issues that should be addressed:

  1. Please add some data regarding main characteristics of JIA. What is the current approach and which are its main disadvantage?

This additional information was originally omitted as there was concern about length of article. We do agree, however, that some brief characteristics for those in the journal’s subscribership that are not familiar with JIA would be beneficial, and have edited our manuscript to reflect this. We have included a JIA Quick facts box, and also a very brief and broad overview of treatment  

2. Authors state that it is challenging to achieve remission in JIA patients. Please add details on remission criteria in JIA. When is patient considered to be in remission.

Similarly to point number 1, whilst originally not included due to concern about length, accepted remission criteria have been added for clarification.

3. It would be valuable to provide diagram for the treatment of JIA (including T2T option), to make it demonstrative for readers.

Given the vast heterogeneity of JIA and treatment options, we felt this would be confusing for the readers, rather than helping. What is standard practice (and even T2T strategy) for one subclass, would differ vastly for another subclass. We offered basic and broad idea of treatment strategies across the subclasses in the main text. 

4. There are some papers regarding the JIA and JIA treatment that have been missed. Please re-check.

Unfortunately, the vast array of papers which look at epidemiology, management and outcomes of JIA are not able to be included in this article, nor is it the main focus. The articles that have been included were intended to be demonstrative of results found in others, given the large registry data. This was to give background of what would be expected in modern management of JIA, and factors that have been found to be predictive of poorer prognosis. We have rechecked and didn't find any newer texts regarding any new texts of JIA T2T since this manuscript was first written.

Round 2

Reviewer 2 Report

The article has somewhat improved in this version. However, a narrative review like this highly benefits if it follows a systematic review writing process.

In both cases above, the articles without supporting references would not move toward scientific evidence-based research. Interpretation can not be one-way, it should be two-way traffic, where you should be able to discuss the consistency, and discrepancy from other studies together with a few lines of your own interpretation.

I am not satisfied with the presentation style of the article. If there is a limitation of the wording, please avoid unnecessary wording and synthesize using the facts.

 The lack of figures/tables/images/schemes in this manuscript makes the article not well presented. The authors have added a few text in the "JIA Quick facts" section which is not self-explanatory. I do not understand why authors want to place that text after the reference is done!

I highly acknowledge the effort the authors have made to bring this work to the scientific community, however, I feel the article has not reached that level for publication yet. So, there needs a major revision.

Author Response

Thank you again for your comments. Please see below as our response. 

The article has somewhat improved in this version. However, a narrative review like this highly benefits if it follows a systematic review writing process.

In both cases above, the articles without supporting references would not move toward scientific evidence-based research. Interpretation can not be one-way, it should be two-way traffic, where you should be able to discuss the consistency, and discrepancy from other studies together with a few lines of your own interpretation.

I am not satisfied with the presentation style of the article. If there is a limitation of the wording, please avoid unnecessary wording and synthesize using the facts.

We acknowledge your dissatisfaction with the style and content of our manuscript. Our primary remit was of a narrative review addressing the title as submitted, and we have framed it to address relevant questions within the topic. This manuscript was reviewed by 3 other peers, who responded more positively to the style and content of the manuscript we have produced.  

The lack of figures/tables/images/schemes in this manuscript makes the article not well presented. The authors have added a few text in the "JIA Quick facts" section which is not self-explanatory. I do not understand why authors want to place that text after the reference is done!

This 'JIA quick facts' box was included at the suggestion of two other reviewers, as many of the subscribers to this journal may not be familiar with this disease. Given there were very few articles of relevance and interest in this area, we didn't feel that a table would value add to this manuscript. We have highlighted the articles of importance in the reference section though, to make these more visible to the reader.   

I highly acknowledge the effort the authors have made to bring this work to the scientific community, however, I feel the article has not reached that level for publication yet. So, there needs a major revision.

We appreciate and respect your views, and thank you again for reviewing our manuscript. We believe that we have met the primary remit from the journal, and received favourable reviews from 3 other peer reviewers. We have re submitted our article as per the last revision for consideration by the editorial team.